# Patterns of ontogenetic evolution across extant marsupials reflect different allometric pathways to ecomorphological diversity

Laura A. B. Wilson ®[1,2] ✉, Camilo López-Aguirre ®[3], Michael Archer[2], Suzanne J. Hand[2], David Flores[4], Fernando Abdala[5] & Norberto P. Giannini ®[5,6]

The relatively high level of morphological diversity in Australasian marsupials compared to that observed among American marsupials remains poorly understood. We undertake a comprehensive macroevolutionary analysis of ontogenetic allometry of American and Australasian marsupials to examine whether the contrasting levels of morphological diversity in these groups are reflected in their patterns of allometric evolution. We collate ontogenetic series for 62 species and 18 families of marsupials ($n = 2091$ specimens), spanning across extant marsupial diversity. Our results demonstrate significant lability of ontogenetic allometric trajectories among American and Australasian marsupials, yet a phylogenetically structured pattern of allometric evolution is preserved. Here we show that species diverging more than 65 million years ago converge in their patterns of ontogenetic allometry under animalivorous and herbivorous diets, and that Australasian marsupials do not show significantly greater variation in patterns of ontogenetic allometry than their American counterparts, despite displaying greater magnitudes of extant ecomorphological diversity.

Understanding how and why some groups of organisms have generated a remarkable amount of morphological diversity whereas others have produced many anatomically and ecologically similar species remains a fundamental challenge that has fascinated both palaeontologists and evolutionary developmental (evo-devo) biologists alike (e.g.,[1–4]). In mammals, a group that shows spectacular adaptive diversity, such contrasts are evident in many lineages[5–8]. One such striking and poorly understood example is the relatively high amount of morphological diversity in Australasian marsupials (Australidelphia), which contrasts with the restricted morphological diversity observed among marsupials in the Americas[9], formerly considered within "Ameridelphia"[10,11], compared here without a rooted tree as partitions of the network of living marsupials. Living marsupials comprise around 6% of modern mammalian diversity, represented by over 400 species, arranged into seven orders. Of these, the four modern orders Dasyuromorphia, Diprotodontia, Notoryctemorphia and Peramelemorphia occur only in mainland Australia, Tasmania, New Guinea and surrounding islands, comprising over 248 species and 18 families[9]. The remaining three orders are represented by over 111 species and 3 families; Microbiotheria, which contains microbiotherian Monito del monte (*Dromiciops gliroides*) as the sole extant form, Paucituberculata, and Didelphimorphia[9]. Although in some cases (e.g., yalkaparidontians[12]), their extinct forms were in Australia[11], the extant species in these groups are distributed in South America and the Neotropics, except the Virginia

[1]School of Archaeology and Anthropology, The Australian National University, Canberra ACT 2600, Australia. [2]Earth & Sustainability Science Research Centre, School of Biological, Earth & Environmental Sciences, University of New South Wales, Sydney, NSW 2052, Australia. [3]Department of Anthropology, University of Toronto Scarborough, Toronto, ON, Canada. [4]Unidad Ejecutora Lillo (Consejo Nacional de Investigaciones Científicas y Técnicas-Fundación Miguel Lillo). Instituto de Vertebrados, Fundación Miguel Lillo. Miguel Lillo 251, CP 4000 Tucumán, Argentina. [5]Unidad Ejecutora Lillo (Consejo Nacional de Investigaciones Científicas y Técnicas-Fundación Miguel Lillo), Miguel Lillo 251, CP4000 Tucumán, Argentina. [6]Cátedra de Biogeografía, Universidad Nacional de Tucumán, Tucumán, Argentina. ✉e-mail: Laura.Wilson@anu.edu.au

opossum (*Didelphis virginiana*), that occupies a range spanning from Central America to southern Canada[11,13].

Since divergence from their relatives in the Americas by the early Eocene[11], marsupials in Australasia have evolved to occupy a broad spectrum of ecological niches and comprise around 40% of modern terrestrial mammal diversity in the region[11,14]. Australasian marsupials show an impressive array of morphological specialisations, having evolved against a backdrop of long-term cycles of climatic changes that led to multiple transitions towards herbivory and grass/browse herbivory, and to occupation of the majority of terrestrial mammalian roles[15,16]. Marsupials in Australasia include forms such as marsupial moles (Notoryctemorphia) that have evolved parasagittal digging locomotion and concomitant specialisations for burrowing in loose soils (e.g., loss of external ears and sight[17]), groups that have exploited carnivorous and insectivorous dietary niches (Dasyuromorphia, forms such as quolls, antechinuses and the Tasmanian devil), and other groups that have adopted omnivorous semi-fossorial specialisations (Peramelemorphia, bandicoots and bilbies). The most diverse order of living marsupials, Diprotodontia, comprises representatives that span a considerable range of body sizes and ecological niches, including browsing and grazing (e.g., kangaroos, koalas, wallabies, wombats) as well as arboreal, nectar feeding species such as the honey possum[18]. Diprotodontian morphological diversity is expanded further among extinct representatives, for example the carnivorous marsupial lions (Thylacoleonidae)[19]. In contrast, in the Americas, extant marsupials are largely restricted to insectivorous, frugivorous and carnivorous dietary niches, with the modern fauna being dominated by opossums[16,20]. Marsupials in the Americas evolved in the presence of multiple herbivorous placental mammal lineages (Xenarthra, native ungulates[21]) that likely reduced capacity to exploit the grazing niche, accounting for the absence of these forms[22,23]. Likewise, New World marsupials evolved side-by-side with a spectacular array of metatherian carnivores in their sister clade, Sparassodonta[24]. South American metatherians consisted of a much broader array of forms during the Cenozoic[22,24]. These included very small (<100 g) to small (<1 kg) insectivorous, frugivorous or omnivorous forms during the early Cenozoic, while in the latter half of this Era, more specialisations appeared, such as granivorous and derived carnivorous forms (e.g., thylacosmilids[22,25–27]).

Dietary diversification influences measures of variability in postnatal cranial growth patterns in some groups, especially examined in rodents[7,28–31], but also uncovered among reptiles (e.g.,[32–34]). These studies highlight how ontogenetic allometry, the relationship between traits and size through ontogeny, may evolve on a macroevolutionary scale with an adaptive base[35], being closely linked to extrinsic factors, such as functional demands associated with dietary specialisation. The macroevolutionary patterning of cranial ontogenetic allometry in extant marsupials remains underexplored, although detailed descriptive studies of individual, or several closely related species, have amassed over the last two decades[36–40]. Several commonalities of postweaning ontogeny have been uncovered for species studied to date, particularly concerning growth of the neurocranium relative to the splanchnocranium[41]. The intra-uterine period of development in marsupials is relatively short, resulting in newborn young that are small and extremely altricial, with most of their skeleton unossified at birth (e.g.,[42,43]). The long extra-uterine development period is accompanied by an antero-posterior gradient of ossification for skeletal elements, represented by early ossification of the oral region as well as accelerated development of the forelimbs for use in climbing, while the hindlimbs remain undifferentiated[5,44,45]. Fundamental shifts from milk-feeding to foods comprising the adult dietary niche are accompanied by growth changes in the skull[18,46], especially growth of the neurocranium relative to the splanchnocranium, to accommodate an increased mass of the temporal muscles[40]. Moderate dietary niche conservatism has been observed among marsupial clades in relation to phylogenetic niche conservatism[47] and the evolution of major dietary patterns (e.g., herbivory) have been shown to reflect phylogenetic structuring[16].

Changes in ontogenetic allometry represent a major pathway to generating morphological variation, creating boundaries to forms realised in phenotypic space. The magnitude and patterning of variation in ontogenetic allometric trajectories may be hypothesised to differ significantly between Australasian and American marsupials considering the contrasting degrees of morphological and dietary diversification in the two groups, reflecting regional differences in faunal composition that have shaped opportunity for dietary niche occupation over time. To test this hypothesis, we collated cranial ontogenetic growth data for 62 species and 18 families of marsupials, covering the entire extant marsupial diversity at the level of all major groups (Fig. 1) and representing the most comprehensive sampling of marsupial postnatal ontogeny to date. We undertake a comprehensive macroevolutionary analysis of ontogenetic allometry by examining the disparity of ontogenetic trajectories (known as allometric disparity, see [48]) at the level of the main partition within the unrooted network of living marsupials (American, Australasian) and Order, as well as disparity associated with dietary habit. We incorporate ontogenetic allometry into a phylogenetic framework to consider the evolutionary patterning of ontogenetic trajectories within and between American and Australasian groups in relation to diet. We assess evolutionary convergence associated with diet and examine whether the magnitude and mode of patterning in ontogenetic allometric morphospace (allometric space) fits common models of trait evolution.

We predict that (1) ontogenetic allometric patterns will show a high level of evolutionary lability in marsupials, based on their eco-morphological diversity; (2) the structuring of allometric space for cranial ontogenetic allometry will differ in magnitude and mode of interspecific variation among Australidelphia compared to American marsupials, with Australidelphia displaying comparatively greater amounts of allometric disparity; (3) the general distribution of species' ontogenetic trajectories in allometric space will reflect phylogenetic structuring; and (4) there will be some evidence for convergence in allometric trajectories among species with similar dietary categories across the two partitions, reflecting patterns observed for ontogenetic trajectories among other mammalian groups, wherein the evolution of ontogenetic allometry has been linked to dietary diversification.

In this work, we show that the magnitude of evolutionary changes in ontogenetic patterns across marsupials is limited, evidenced by species' distribution in allometric space and few inter-specific differences in allometric slope, especially among members of Australidelphia. We recover diet-specific patterns of convergence in ontogenetic trajectories between Australasian and American marsupials, indicating that convergences in adult cranial morphology arise from similar ontogenetic trajectories.

## Results

### Ontogenetic allometry patterns for Australidelphia and "Ameridelphia", among orders and dietary categories

Initial homogeneity of slopes (HOS) testing, to assess species-level variation of trajectories within groups, revealed significant differences in slope among species within each dietary group ($P < 0.0001$–0.03, Table S1), within "Ameridelphia" (Likelihood ratio statistic: 239.6, $P < 0.0001$), and within Australidelphia (Likelihood ratio statistic = 244.1, $P < 0.0001$) (Table S1). Examination of pairwise comparisons across all possible pairs of species in the sample ($n = 1109$) revealed that most (89 %, $n = 978$) species did not differ significantly in slope to one another. Among the comparisons that presented significantly different slopes ($n = 131$), most were located within Australidelphia ($n = 84$) compared to "Ameridelphia" ($n = 47$), but the proportion of significant differences was over six times greater for "Ameridelphia" (47/91 comparisons, 51%) compared to Australidelphia (84/1089 comparisons, 8%). Within dietary groups,

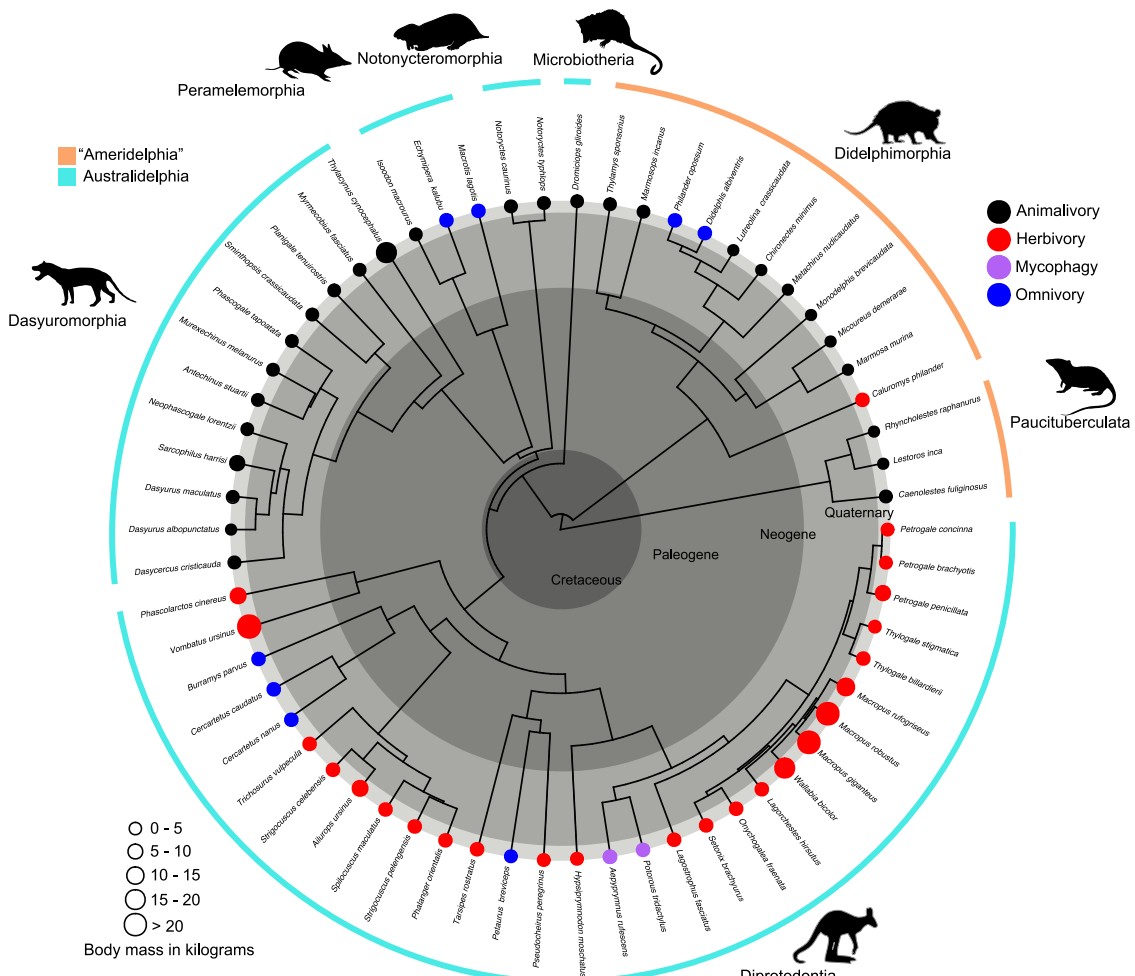

**Fig. 1 | Overview of the time-calibrated phylogenetic relationships and average adult body mass for species sampled in this study.** Dietary habits are shown with coloured symbols representing animalivory (black), herbivory (red), mycophagy (purple) and omnivory (blue). The sample comprised cranial ontogenetic series for 14 species belonging to partition "Ameridelphia" (orange shade) and 48 species belonging to partition Australidelphia (light blue shade) (*n* = 2091 specimens). Animal silhouettes sourced from PhyloPic (www.phylopic.org). Source data are provided as a Source Data file.

the greatest number of significant pairwise differences in slope were found among animalivorous species (65/351 comparisons), followed by herbivorous species (21/276) and omnivorous species (8/28). Species with omnivorous diets demonstrated the greatest proportion of significant within-group differences in slope (29%) (Fig. 2). All slope values are provided in the Source Data File and associated data.

The homogeneity of slopes (HOS) test rejected the null hypothesis of equal slopes (Likelihood ratio statistic: 37.43, *P* < 0.0001) and the null hypothesis of equal intercepts (Wald statistic = 64.24, *P* < 0.0001) for pooled comparison of allometric trajectories for Australidelphia (*n* = 1389) compared to "Ameridelphia" (*n* = 702) (Table S2). Among marsupial orders (*n* = 7), the HOS tests revealed significant differences in both slope (Likelihood ratio statistic = 278.1, *P* < 0.0001) and intercept (Wald statistic = 246, *P* < 0.0001) (Table S2). Pair-wise comparisons among orders indicated that, after correction for multiple comparisons, over half of pairs showed significant differences in slope (12/21 comparisons) and in intercept (15/21 comparisons) (Supplementary Table 3). Pairwise comparisons that yielded similar slopes and intercepts were notably found for Microbiotheria, Notoryctemorphia, Paucituberculata and Peramelemorphia. Statistically significant differences were recovered among pooled trajectories for dietary categories (Fig. 3a), with significant slope (Likelihood ratio statistic = 53.24, *P* < 0.0001) and intercept (Wald statistic = 186, *P* < 0.0001) differences being present among most groups. The exceptions were, similar slopes for omnivores compared to both animalivores (*P* = 0.62) and

herbivores (*P* = 0.99), and similar intercepts for herbivores and myco-phagous species (*P* = 0.99) (Supplementary Table 4). Comparing trajectories using the interaction between partition ("Ameridelphia", Australidelphia) and diet, to assess potential convergence in trajectories among dietary groups across the two partitions, revealed significant differences in slope for most (15/21) pairs, with notable exceptions among animalivores and omnivores in both partitions sharing similar slopes (Fig. 2, Fig. 3a). Most pairs also differed significantly in intercept (15/21 comparisons) (Supplementary Table 5).

**Allometric space for Australidelphia and "Ameridelphia"**
Allometric space was constructed to examine the magnitude and patterning of variation in ontogenetic trajectories among Australasian and American marsupials. Ordination of principal component one (PC1, 44.7%) and principal component two (PC2, 14.7%) for allometric space captured a total of 59.4% of variance across species' ontogenetic trajectories (Supplementary Figure 1) and revealed that most species belonging to "Ameridelphia" and Australiadelphia fall within a shared region of allometric space (Fig. 3b). Considerable expansion of species' occupation of morphospace was evident along PC1, owing to the position of the spotted-tail quoll (*Dasyurus maculatus*), being located at the extreme positive end of PC1, along with the rufous bettong (*Aepyprymnus rufescens*) (positive PC1 and PC2 scores), and grey slender opossum (*Marmosops incanus*). The extreme negative portion of PC1 was occupied by the Tasmanian devil (*Sarcophilus harrisii*)

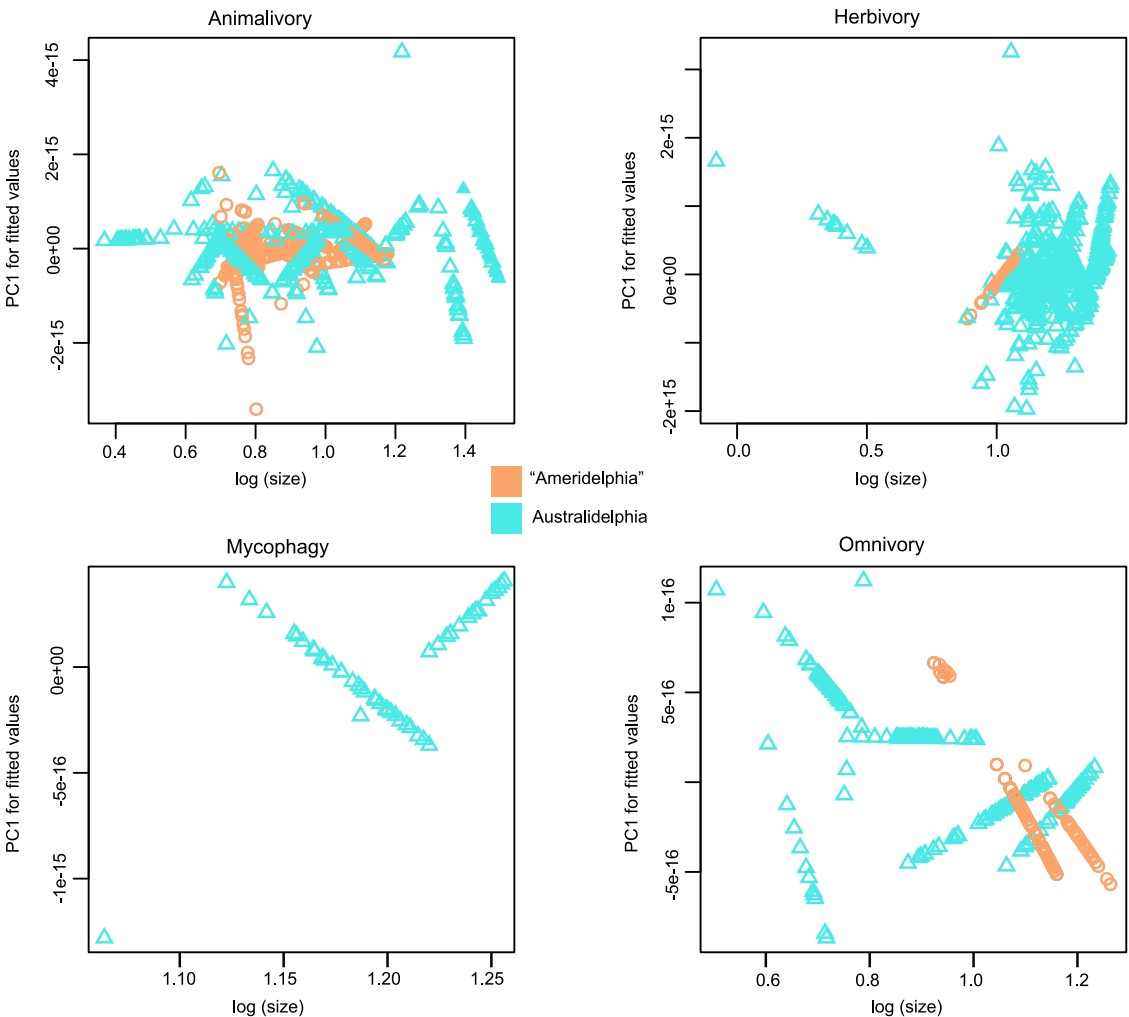

**Fig. 2 | Ontogenetic allometric trajectories for marsupials sampled in this study.** The sample comprised cranial ontogenetic series for 14 species belonging to partition "Ameridelphia" (orange circle) and 48 species belonging to partition Australidelphia (blue triangle) (*n* = 2091 specimens) grouped into four dietary categories. Raw trajectories are derived from plotting log transformed geometric mean (log(size)) against PC1 of the predicted values of multivariate regression of shape ratios on size (PC1 for fitted values). Source data are provided as a Source Data file.

(PC1 = −5.0) (Fig. 3b and Supplementary Fig. 2). Moderate expansion of species' occupation along the positive end of PC2 ( > 2.0) resulted from the position of rufous bettong *Aepyprymnus rufescens*, and to a slightly lesser extent by the common spotted cuscus (*Spilocuscus maculatus*). Since allometric space represents the amount of divergence between ontogenetic allometric trajectories in relation to the mean ontogenetic allometric trajectory (located at 0,0), positive and negative PC loadings for a given trait represent faster or slower than average growth for that trait. Here, the traits with the greatest effect (largest positive/negative loading) on PC1 corresponded to faster than average growth of palatal length (LPAL) (0.13), along with slower than average growth of height of the mandibular body (HD) (−0.37) and height of the muzzle (HM) (−0.34) (Supplementary Table 6). PC2 was associated with faster than average growth of length of upper postcanine row (Upos) (0.31) and height of the muzzle (HM) (0.17), in addition to slower than average growth of orbital length (ORB) (−0.54) and LPAL (−0.45) (Supplementary Table 5).

## Morphological disparity measures
Morphological disparity was calculated for phylogenetic and dietary groupings to assess the magnitude of variation in slope across species in allometric space. Morphological disparity calculated from PC scores extracted from allometric space revealed slightly higher Procrustes Variance (PV) values for Australidelphia (PV = 13.0) compared to

"Ameridelphia" (PV = 10.2), though pairwise differences were not significant (*P* = 0.77) (Supplementary Table 7) (Fig. 4). Similarly, pairwise comparisons of morphological disparity among dietary groups did not reveal significant differences. Among dietary groups, the greatest values for PV were observed among mycophagous taxa (PV = 45.9), whereas omnivorous taxa showed the smallest magnitudes of morphological disparity (PV = 4.7) (Supplementary Table 7 and Fig. 4a). Across Orders, morphological disparity was highest for members of Dasyuromorphia (PV = 20.8), and lowest among members of Microbiotheria (PV = 3.7) and Paucituberculata (PV = 3.0) (Supplementary Table 8 and Fig. 4b), though pairwise comparisons did not reveal significant differences between any pairs. For groups comprising the interaction between partition ("Ameridelphia", Australidelphia) and diet, the greatest values for disparity were found for mycophagous species belonging to Australidelphia, represented in the sample by the long-nosed potoroo (*Potorous tridactylus*) and rufous bettong (*Aepyprymnus rufescens*) (PV = 45.9), whereas disparity was lowest for omnivorous taxa in both "Ameridelphia" (PV = 4.3) and Australidelphia (PV = 5.0) (Supplementary Table 9).

## Evolution of ontogenetic allometric trajectories among clades and dietary groups
Convergence tests were conducted to assess the extent to which species belonging to the same dietary group displayed convergence in

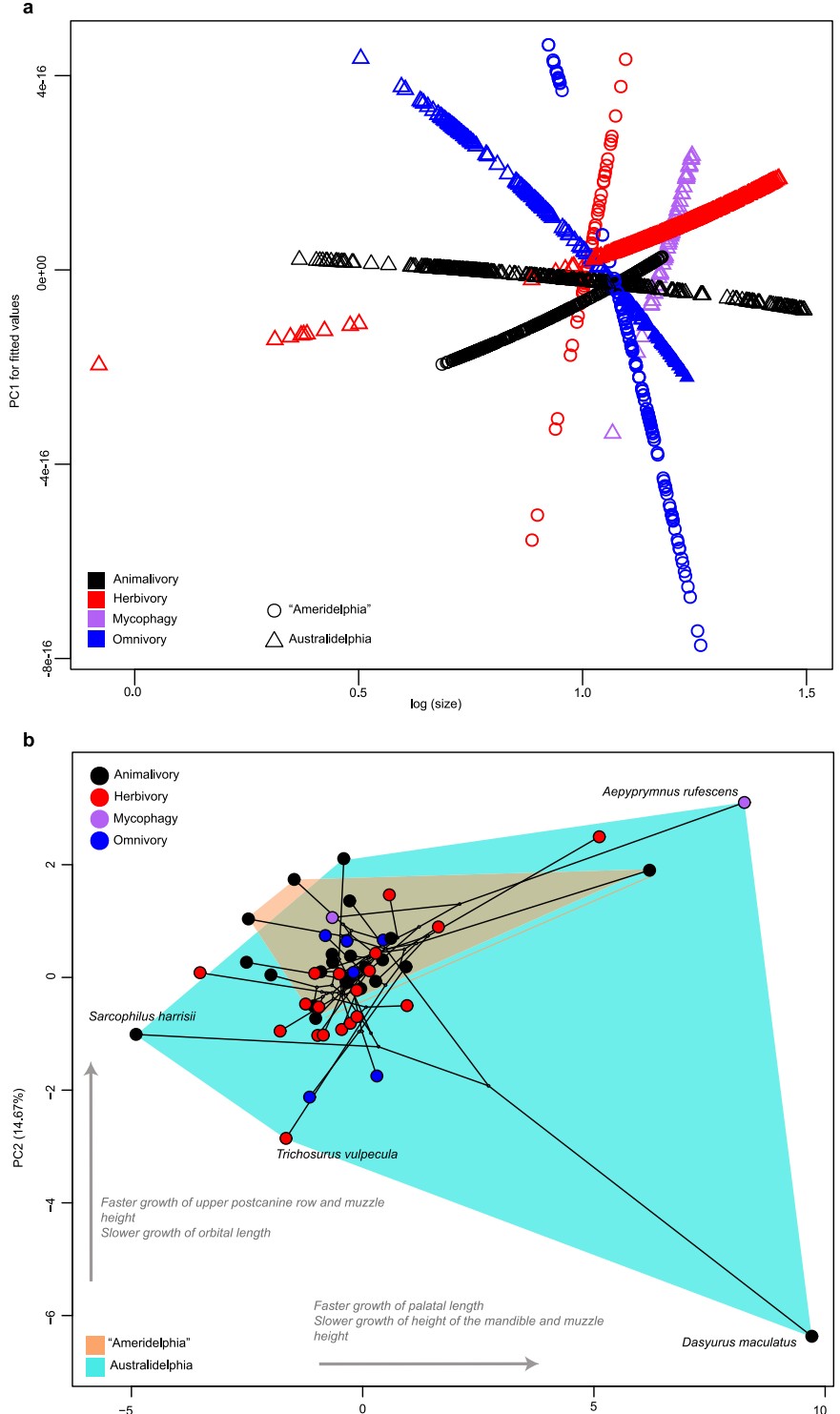

**Fig. 3 | Allometric trajectories and allometric space. a** Pooled ontogenetic allometric trajectories grouped by partition, represented by "Ameridelphia" (circle) and Australidelphia (triangle), and by dietary habit, comprising animalivory (black), herbivory (red), mycophagy (purple) and omnivory (blue) groups. **b** Allometric space for marsupials, with time-calibrated phylogeny projected onto species distributions. Each point in allometric space represents the ontogenetic allometric trajectory for an individual species, and shape changes along the Principal Component (PC) axes represent faster/slower than average growth for given features, noted next to arrows for PC1 and PC2. Source data are provided as a Source Data file.

ontogenetic allometry between Australidelphia and "Ameridelphia". Results of the convergence tests that applied C1-C4 distance-based metrics based on species' location in allometric space (i.e., PC score), revealed significant convergence between Australasian and American representatives for both animalivory and herbivory groups

(Supplementary Table 10). C2 values were notably higher, indicating greater convergence, for animalivores (C2 = 0.631) and for herbivores (C2 = 0.818) in comparison to omnivores (C2 = 0.114) (Supplementary Table 10). The convergence tests that adopted angular comparisons between representatives from Australidelphia and those from

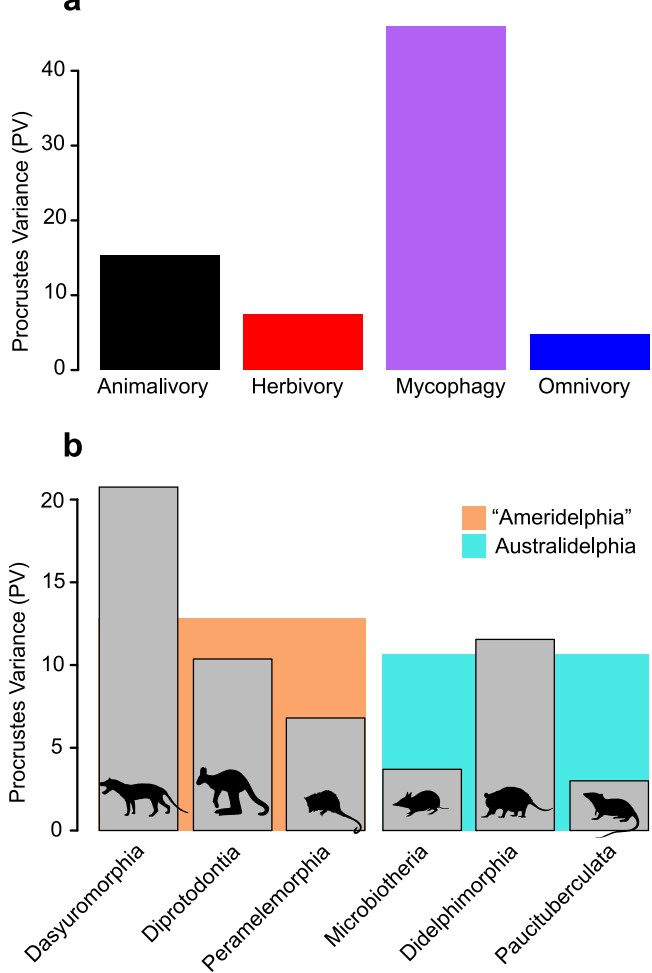

**Fig. 4 | Morphological disparity values. a** Disparity values (Procrustes Variance) shown for species ontogenetic trajectories pooled into dietary categories, represented by animalivory (black), herbivory (red), mycophagy (purple) and omnivory (blue). **b** Disparity values shown for Orders (grey shaded bars), and for "Ameridelphia" and Australidelphia partitions (orange and blue shaded bars, respectively). Animal silhouettes sourced from PhyloPic (www.phylopic.org). Source data are provided as a Source Data file.

"Ameridelphia" belonging to animalivory and omnivory dietary categories (as states) revealed large angular differences between the clades (Supplementary Table 11). Most mean angular comparisons were not significant, ranging between 65 and 92 degrees (Supplementary Table 11), with the exception of convergence among animalivorous species in Australidelphia and "Ameridelphia" ($P = 0.03$). Adjusting these angular values by time distance between species' pair tips (ang.state.time) resulted in much smaller angle values owing to the large time differences across the tree (0.4–0.5 degrees). Both convergence methods supported strong convergence among animalivorous species and absence of convergence among omnivorous species.

### Evolutionary modelling

We conducted a series of model fit assessments to identify whether differences in modes of evolution across diets and partitions ("Ameridelphia" and Australidelphia) were present. Our model fitting results, comparing the fit of 12 competing models of evolution to assess differences in modes of evolution across diets and partitions ("Ameridelphia" and Australidelphia), indicated that an Ornstein-Uhlenbeck (OU) model best fit the distribution of species in allometric space (Supplementary Table 11). Model fit (AICc) among the OU models

ranged from 771.7 to 735.9, compared to AICc values of 816.8 for an Early Burst model and between 962.5 and 774.0 for Brownian Motion models (Table S11). The only model meeting the best supported criterion of a delta AICc (dAICc) value of below 2.0 was the OU1 model (dAICc = 0), which assumed evolution under a single optimum. The next best fitting models were OU models that assumed different optima based on diet (OUd, dAICc = 5.5) and partitions (OUs, dAICc = 9.0; Supplementary Table 12).

## Discussion

The magnitude of ecomorphological diversity in marsupials has been paralleled, but also contrasted, with morphological diversity evidenced by placental mammals (e.g.,[8]). As a major factor shaping trait variation, patterns of ontogenetic allometry may be highly conserved or variable under scenarios of low or high morphological diversity, respectively. Despite contrasting levels of morphological diversity among Australasian and American groups, allometric space for marsupials presents some phylogenetic structuring, with trajectory evolution best fitting a stabilising (single-optimum Ornstein-Uhlenbeck) model of evolution. The magnitude of allometric disparity, that is the amount of variability in slopes among species' allometric trajectories, is slightly greater in Australasian marsupials compared to their less morphologically diverse counterparts in the Americas, but not to a significant degree.

A phylogenetically-constrained structuring of allometric space is further reflected through pairwise comparisons of allometric slopes across all species in the sample, which indicated that most slopes did not differ significantly from one another (89%), in contrast to findings for other mammalian groups, albeit generated from samples that captured comparatively lower amounts of species diversity (e.g.,[28]). Focusing on within-group comparisons of allometric slopes, consistent with predictions that ontogenetic trajectories will be more variable among the more morphologically diverse Australidelphia, the latter partition contained a greater number of significant pairwise differences compared to American species. However, despite appearing less morphological diverse than their Australasian counterparts, American marsupials show a much higher proportion of within-group variation in allometric slope, consistent with the small magnitude of difference in allometric disparity between the two partitions (see Fig. 4).

Our results evidence changes in ontogenetic slope and intercept as common at deeper (clade) nodes, consistent with evolutionary lability of postnatal ontogenetic allometric patterns between Australasian and American groups. Robust differences in trajectories between Australasian and American groups and among members with similar diets, coupled with strong evidence of convergent evolution associated with animalivory, in particular, highlight a complex narrative for ontogenetic evolution in marsupials. Species that diverged more than 65 million years ago have adapted to the same dietary niches and show evidence of convergence in ontogenetic allometric trajectory, echoing ontogenetic evolution patterns associated with diet that have been observed among other mammals[28].

### Limited occupation of allometric space

We predicted that variability in ontogenetic trajectories would broadly reflect the underlying differences in adult morphological diversity among "Ameridelphia" and Australidelphia; however, our results did not yield strong evidence for this assertion. We did not recover significant differences in allometric disparity, the extent to which taxa occupied disparate regions of allometric space, among Orders, partitions ("Ameridelphia", Australidelphia), dietary habit, or interactions of the latter two (partition × diet). Together these results indicate that, with the exception of several outlier species that contributed to many of the significant species-level pairwise comparisons, the occupation of allometric space across marsupials appears limited. Notably, the

species expanding occupation of allometric space were mostly Australidelphians, represented by several dasyurids (spotted-tailed quoll *Dasyurus maculatus*, Tasmanian devil *Sarcophilus harrisii*), plus the common spotted cuscus (*Spilocuscus maculatus*) and rufous bettong (*Aepyprymnus rufescens*), whereas the Gray slender opossum (*Marmosops incanus*) was the only representative from the Americas (Fig. 3 and Supplementary Table 2), though many species belonging to "Ameridelphia" exhibited significant pairwise differences in slope. The spotted-tail quoll and Tasmanian devil both have an especially short rostrum with no evidence for a third premolar being present, including an absence of deciduous precursor for the tooth in juveniles of both species[11], which is reflected in the strong impact of mandibular body height and muzzle height growth on PC1 of allometric space (Supplementary Table 5), likely explaining the extreme positive excursion along this axis by the spotted-tailed quoll (Supplementary Fig. 3). Similarly, PC2 captured faster than average growth of the upper postcanine tooth row, therefore negative values along this axis, as exhibited by both aforementioned dasyurids, would further contribute to a shorter rostrum relative to other species in the sample. The trends on PC1 of positive allometry for length of the palate and negative allometry for height of the muzzle, oppose those on PC2 of negative allometry for length of the palate and positive allometry for height of the muzzle, and together may explain the disparate positions of the spotted-tailed quoll and Tasmanian devil in allometric space, with these variables contributing to attainment of a short face.

Consistent with our findings of relatively restricted allometric space occupation, marsupials have been considered as constrained in their evolution due to their unique reproductive mode[5,45,49], though others have disagreed on the grounds of limited data[50], and geographic patterns have been underappreciated as a potential driver of these differences[8]. The prenatal period is relatively short in marsupials, resulting in newborn young that are small and extremely altricial, with most of their skeleton unossified at birth (e.g.,[42,43]). Newborns continue their development externally, undertaking a crawl to the mother's pouch to suckle, a feat that is supported by early ossification of the oral region as well as accelerated development of the forelimbs for use in climbing[5,44,45]. This event sequence contrasts that seen for placentals, which, for example, show early ossification of cranial roof elements[51], and has been proposed to have shaped marsupial morphological diversity. Further, the absence of features such as flippers and wings has been linked to the unique perinatal biology of marsupials[52–54], though acknowledging that marsupials have many craniodental features that appear derived compared to placentals (e.g.,[55]). Cranial evolution in marsupials has been shown to exhibit limited capacity for evolutionary flexibility[56–58], a feature attributed to high magnitudes of integration, a measure of connectedness among morphological traits[59]. Studies of both adult[60] and prenatal crania[61], though with limited species sampling, have evidenced particularly strong integration of the oral apparatus, citing the unique demands for continuous suckling[62]. Notably, peramelamorphians have been highlighted as potentially released from a highly integrated state for the oral apparatus[60], because they evolved a chorioallantoic placenta, convergent with that of placentals, such that they do not undertake a crawl to the pouch[52]. Consistent with this suggestion, peramelamorphians extend adult cranial morphospace occupation relative to other marsupials, increasing disparity at the macro-level[60]. Similarly, here, peramelamorphians yield similar slopes and intercepts as compared to Microbiotheria, Notoryctemorphia, and Paucituberculata, however we also find several species (Greater bilby *Macrotis lagotis*, Common spiny bandicoot *Echymipera kalubu*) occupy a distinct position along the negative (−2.0) portion of PC2 in allometric space (Supplementary Fig. 2).

Under conditions of high magnitudes of integration, morphological disparity is achieved mostly by size changes in response to selection pressures, making allometry a crucial generator of morphological variation in marsupials[56–58,63]. Magnitudes of integration are inter-related to life history parameters in mammals[64]. In contrast to mammals that give birth to precocial young (e.g., bovids), high levels of energy investment in growth of altricial young yield significant magnitudes of size variation across lineages and high magnitudes of cranial integration. This is observed in marsupials[64] and likely echoed by the phylogenetic structuring of differences in allometric parameters evidenced here among Orders but absent for comparisons across dietary categories. Size-related disparity has been uncovered as clade-specific in didelphids[65], and present in other morphological traits in response to ecological pressures[66–68].

## Dietary diversification and convergence in ontogenetic allometric patterns

We hypothesised that evolution of allometric trajectories among marsupials may follow an adaptive basis, similar to that of other altricial-young bearing mammals (rodents[28,30]), and that dietary habit may explain variation in patterning of allometric space. Between American and Australasian marsupials, we found evidence of convergence in ontogenetic allometric trajectories associated with animalivory, and, to a lesser extent, with herbivory. However, magnitudes of allometric disparity among dietary groups did not differ significantly, with allometric space showing limited segregation for animalivorous and omnivorous taxa (Fig. 3). Notably, the two dietary groups that presented as significantly convergent among Australasian and American taxa, reveal contrasting magnitudes of inter-specific trajectory variability, whereby animalivores show a greater proportion of significant differences in slope compared to slope differences present among herbivores (Fig. 2), which are largely homogenous in their allometric patterns. Consistent with the convergence in ontogenetic allometry recovered here among animalivores, convergence in adult cranial morphology has previously been identified among animalivorous species of small didelphids and small dasyuruids present in our sample[20], suggesting that the convergences in allometric trajectories recovered herein represent pathways to similar adult morphology. Convergence in adult cranial morphology between the New Guinean quoll (*Dasyurus albopunctatus*) and the big lutrine opossum (*Lutreolina crassicaudata*) has especially highlighted shared short, strong rostra among these species[20], features that also contribute to PC1 of allometric space.

Among the 12 evolutionary models tested, an OU model with a single optimum and an OU model with different optima based on diet (Supplementary Table 12) were best and second best supported, followed by an OU model with different optima for Australasian and American partitions. These results indicate that dietary habit has likely independently shaped ontogenetic evolution among marsupials on both continents, to differing degrees, which is consistent with differences in the composition and availability of vacant niches, and connections that have been drawn between extant adult cranial diversity and large-scale palaeo-climatic events, in addition to concurrent diversification of placental lineages[16]. Especially, marsupials in the Americas lack extreme feeding behaviour specialisations (e.g., myrmecophagy)[24,69], and their overall generalised morphology has been hypothesised as sufficient to meet feeding requirements of a predominantly omnivore/insectivore dietary pattern[68], noting Calyuromyinae (woolly opossums and relatives) are unique among extant American lineages in evolving predominant herbivory[16]. A clade-wide evaluation of cranial shape evolution among didelphids similarly recovered a lack of inter-specific shape differences associated with diet[68], finding strongly conserved phylogenetic structuring to cranial morphospace, also replicated in narrower evolutionary comparisons among members of the group[63]. We further note that an evolutionary shift to browse herbivory occurred only once in marsupials, in Diprotodontia (Fig. 1)[16]. Although herbivorous taxa overlap more-or-less completely with animalivorous and omnivorous taxa in allometric space, indicating a lack of shift in the patterning of cranial shape-size

covariance associated with unlocking this novel niche (Fig. 3), we find muzzle and mandibular height as significant contributors to PC1 of allometric space, and species-level trajectories among herbivores are mainly concordant in their patterning (see Fig. 2). That interspecific shifts in ontogenetic allometry of the rostral region account for major portions of variance along PC1 is consistent with the biomechanical importance of shortening the rostrum to enable greater bite force via a shorter out-lever, relevant in processing of resistant vegetation in diprotodontians[70] as well as the bone-crushing capabilities of the spotted-tailed quoll (*Dasyurus maculatus*)[71]. As noted earlier[20], patterning of skull growth may not reflect differences in diet in full, perhaps only after dentition characteristics (not accounted by allometric growth) are factored in.

Among the many, well-known instances of convergence, which serve to highlight how marsupials have evolved similar solutions to specialisations seen in placentals[72,73], only broad similarity between the extinct marsupial thylacine and the placental grey wolf has been examined within a framework of ontogenetic allometric trajectory comparison[74]. Other examples, such as the marsupial mole sharing similar digging adaptations with placental moles, the numbat representing a marsupial form of anteater, and the Tasmanian devil and quolls sharing similarities with placental carnivores[75,76], reveal the prevalence of functional equivalencies despite differing developmental strategies and integration magnitudes[62], although the ontogenetic basis for these similarities remains opaque. In the case of the placental grey wolf and the thylacine, which are superficially similar in the broad context of body form as dog-sized carnivores, parallel trajectories were uncovered over the course of ontogeny. Convergent patterns of postnatal ontogeny among "Ameridelphian" and Australidelphian members with the same diet were most strongly supported herein for animalivory, and to a lesser extent for herbivory, though results in the case of herbivory are tentative owing to the small number of herbivorous taxa in "Ameridelphia". Nevertheless, the substantial differences in allometric trajectories between species with the same diets from the Americas and Australasia (Supplementary Table 5), highlight the evolution of differing ontogenetic pathways, independent of dietary similarity.

The evolutionary patterning of ontogenetic trajectories captures how the changes in size and proportions of body parts that underpin the evolution of form, and thus phenotype, are patterned in space and time. Using the most comprehensive analysis of ontogenetic evolution in marsupial mammals to date, our findings support a complex narrative of continent-specific factors shaping the evolution of ontogenetic allometric trajectories in marsupials. It is evident that the magnitude of evolutionary changes in ontogenetic patterns across marsupials is limited, shown in occupation of allometric space and few inter-specific differences in allometric slope, especially among members of Australidelphia. We recover diet-specific patterns of convergence in ontogenetic trajectories between Australasian and American marsupials, indicating that previously identified convergences in adult cranial morphology associated with animalivory arise from similar ontogenetic trajectories. Lability in ontogenetic patterns present at deeper nodes, and among omnivores in particular, may reflect incomplete convergence, shaped by regional differences in faunal composition.

## Methods
### Study sample
We analysed 2091 specimens representing ontogenetic series of 62 species, comprising coverage of 18 families of marsupials including members of all seven living Orders. Average adult body mass ranged across five orders of magnitude among species in the sample, from 4 grams (g) (*Planigale tenuirostris*, narrow-nosed planigale) to >39 kilograms (kg) (*Macropus giganteus*, eastern grey kangaroo). Among the more speciose groups, average adult body mass ranged from 4 g to

16.7 kg in Dasyuromorphia, from 19 g to 1.3 kg in Didelphimorphia and from 9 g to 39 kg in Diprotodontia (Fig. 1). Raw ontogenetic measurement data (ontogenetic series) were compiled for 43 marsupial species (1721 specimens[41]) along with published raw measurements (see[77]), and the total data set was used in a completely novel and distinct set of analyses, assessing allometric evolution. Cranial measurement data were collected from institutions in North and South America, and Australia, as follows: Australian Museum, Sydney (AM), American Museum of Natural History, New York (AMNH), Centro Nacional Patagónico, Puerto Madryn (CNP), Field Museum of Natural History, Chicago (FMNH), Museo Argentino de Ciencias Naturales Bernardo Rivadavia, Buenos Aires (MACN), Museo de La Plata, La Plata (MLP), Museu Nacional Universidade Federal do Rio de Janeiro, Rio De Janeiro (MNRIO), Museu de Zoologia da Universidade de São Paulo, São Paulo (MZUSP), and Western Australian Museum, Perth (WAM). Following previous studies[38–40], the measurements comprised a set of 14 variables, including both both neurocranial and splanchnocranial variables and accommodating all the major dimensions of the skull. These were: BB, breadth of braincase; BPAL, breadth of palate; CBL, condylobasal length; HD, height of mandibular body; HM, height of muzzle; LC, length of coronoid process; LD, length of dentary; LN, length of nasals; LPAL, length of palate; LPos, length of lower postcanine row; OH, height of occipital plate; ORB, length of orbit; PAL, length of palate; UPos, length of upper postcanine row; ZB, zygomatic breadth.

### Phylogenetic framework
To enable examination of evolutionary diversification of allometric coefficients, we adopted the maximum likelihood phylogeny of Mitchell et al.[78], following our previous study[20]. The phylogeny comprises 97% of extant marsupial diversity at the genus level, and 58% of diversity at the species level. Here and recently[20], we selected this phylogeny, which is based on a supermatrix approach, as it offers robust fossil calibration using 14 fossil data points and, further, is consistent with the current consensus of inter-ordinal relationships[9]. We supplemented taxonomic coverage, for several groups not covered by Mitchell et al.[78] and locally adjusted branch lengths for inter-relationships within Caenolestidae, extracted from Ojala-Barbour et al.[79] and for didelphids, taken from Amador and Giannini[65].

### Dietary categories
Each species was assigned a dietary category, following the comprehensive assessment of marsupial diet provided in Amador and Giannini[16], which collated available data on natural diet, taken from faecal contents, stomach contents or direct behavioural observations, for each of the 193 species represented in the marsupial phylogeny presented by Mitchell et al.[78]. Dietary categories followed the simplified four-state dietary scheme created by Amador and Giannini[16], represented as animalivory, herbivory, mycophagy and omnivory. This classification system was created by procuring information from the literature on natural diet, using a minimum of five reference sources per taxon, and assignments for the four-state scheme were based on predominant food source in the diet (>50% frequency of consumption)[16].

### Ontogenetic allometry trajectories
To conduct allometric regressions of cranial shape ~ cranial size, the cranial measurements were transformed to Log-shape ratios[80]. The use of log-shape ratios requires that a standard size variable is computed to represent the overall size of the object, and shape is then quantified as a vector of shape ratios[80,81]. For each specimen, size was computed as the geometric mean of all 14 cranial measurements, and each measurement was then divided by size to produce a shape ratio. The shape ratios were log-transformed and then used as the raw data for subsequent analyses.

Using 62 species (*n* = 2091), we performed standardised major axis (SMA) regressions with the 14 morphometric measurements, as

implemented in the function *sma* from the R package smatr version 3.4.8[82]. SMA models were tested against the null hypothesis of isometry[83]. SMA regressions comprise a homogeneity of slopes (HOS) test to assess, using the Likelihood Ratio statistic[83], the patterning of allometric relationships (i.e., the direction and magnitude of morphological change with size) including interaction terms, detailed as follows. In a phylogenetic context, contrasts were made first for the grouping variables of partitions, to assess differences between "Ameridelphia" and Australidelphia. "Ameridelphia" is paraphyletic in recent marsupial phylogenies rooted in a non-marsupial (e.g., stem metatherian) outgroup (e.g.,[11]); however, as used here without outgroup, "Ameridelphia" becomes one partition of the unrooted network of living marsupials, opposing the complementary partition Australidelphia (see Giannini[84] for a general case). Next, we tested differences at the Order level, to assess differences across all 7 Orders. The null hypothesis of equal slopes and intercepts (shape ~ size) were compared against models with the following interaction terms: partition ("Ameridelphia", Australidelphia) (shape ~ size × partition), Order (shape ~ size × Order), and diet (shape ~ size × diet). Pairwise comparisons of slope and intercept were performed for partitions, Orders and dietary categories using the multcomp function and *P* values were adjusted for multiple comparisons using a Šidák correction[85]. To assess whether American and Australasian marsupials with similar diets converged in allometric trajectories, SMA regressions were run including a partition × diet interaction term and pairwise comparisons were done using the protocol described above. These regressions each involved the pooling of ontogenetic trajectories for dietary habit, partition and Order. Prior to pooling species ontogenetic trajectories into groups, we undertook HOS tests to assess species' level differences in slope, comprising a total of 1109 comparisons. Among these, only 131 pairwise comparisons showed significant differences in slope (11%). Therefore, we focus presentation of our results and discussion on outputs of the pooled comparisons, and note the species-level differences in slope throughout, providing all raw slope data on Zenodo[86].

Out of the total 62 species, the dataset was then pruned to comprise 50 species ($n = 1949$), which were represented by at least 15 specimens, i.e., a greater number of specimens than shape ratio variables ($n = 14$). All downstream analyses were conducted separately with this pruned dataset ($n = 50$ species), and were based on allometric coefficients extracted from individual species regressions of size ~ shape ratios, which were performed using the R base linear model function *lm()*.

The first principal component (PC1) of the predicted values extracted from the regression of shape ratios on size was used to visualise ontogenetic allometric trajectories. Allometric space was constructed by performing a Principal Component Analysis (PCA) on the allometric coefficients of the species-level individual regressions ($n = 50$ species), representing the ontogenetic allometric trajectory, for each species. For each species, the allometric coefficients represent the growth trends for each measurement, as assessed across all specimens in the sample. As an example, for the opossum *Caluromys philander* (represented by 40 specimens), the multivariate regression of shape variables ($n = 14$) ~ size performed on 40 specimens, produces a set of 14 allometric coefficients, capturing the ontogenetic allometric trajectory. These 14 coefficients were compiled for each species ($n = 50$) in the pruned sample ($n = 1949$ specimens) and used as input to the PCA (50 taxa × 14 allometric coefficients). As such, each point in allometric space represents the ontogenetic trajectory of a species (see[7,87,88] for similar approach). Allometric space ordinations enable visualisation of the patterning and magnitude of interspecific variation in allometric trajectories, and the principal component (PC) scores can be used to quantify disparity in allometric space[48].

Morphological disparity was calculated for phylogenetic (partition, Order) and dietary groups using both scores extracted from allometric space projections (PC scores) and residuals from the HOS models, which represent an allometric-corrected disparity measure. The use of residuals in the context of ontogenetic allometry data provides an estimate of allometry-corrected dispersion of the data along the trajectory (see[87] for a similar approach). Morphological disparity calculations were performed using the function morphol.disparity in R package geomorph, which estimates variance (hereafter referred to as Procrustes variance [PV]). PV is the trace of the group covariance matrix, here divided by number of species in the group, therefore accounting for unequal group sizes. Absolute differences in PV were used to test differences in morphological disparity among groups[89]. The statistical significance of pairwise distances between groups was assessed using a permutation test (10,000 iterations).

## Evolution of ontogenetic allometric trajectories among clades and dietary groups

To assess whether the American and Australasian partitions show evidence of convergent evolution in allometric trajectories associated with dietary ecology, convergence tests were performed using the function convratsig of the R package convevol version 1.3[90]. Specifically, we tested for allometric convergence between Ameridelphian and Australidelphian species with similar diets. Statistical significance was assessed separately for taxa classified as animalivores, herbivores, and omnivores using 300 simulations across the phylogeny. Runs were completed at 100, 200, 300 and 1000 simulations, and results for >200 simulations appeared highly similar (<0.0003 difference in C value). The mycophagy dietary group was excluded as it was not represented in the "Ameridelphia" sample. The first four principal components summarising 76.12% of variation in allometric trajectories were used as input data, which represented principal components comprising significant proportions of variation in the sample, as determined through a broken stick model[91], and the maximum matrix dimensions executable on a standard workstation (Supplementary Figure 1). This convergence test uses four distance-based indices (C1–C4) that compare the magnitude of convergence within (C1, C2) and between (C3, C4) lineages[90]. C index values measure convergence on an increasing scale, such that zero reflects an absence of convergence and larger values denote greater convergence. C1, ranging between 0 and 1, measures the phenotypic distance between convergent tips relative to the maximum evolutionary distance between two lineages, reflecting how much subsequent evolution has reduced inter-lineage distance. C2 assesses the magnitude of evolution that has occurred as a result of convergence, with larger values indicating greater amounts of convergence. Similar to C1, C3 and C4 reflect proportions such that C3 indicates how much convergence has occurred as a proportion of the total amount of evolution along lineages leading from their common ancestor and C4 indicates how much convergence has occurred relative to the entire clade, defined by the common ancestor of convergent taxa.

To corroborate these results and assess the sensitivity of our results, we implemented a further convergence test using the search.conv function of the R package RRphylo version 2.5.8[92,93]. We applied the same protocol described above, using as input data both the species-level allometric coefficients and the scores of the PCA. Statistical significance was assessed using 300 simulations. This convergence test adopted a within/between states (here, dietary categories) approach, whereby the mean angle between all possible species pairs (i.e., Ameridelphian animalivore – Australidelphian animalivore, Ameridelphian omnivore – Australidelphian omnivore) evolving under a dietary category (state) is returned (ang.state), plus the mean angle divided by time distance between respective tips (ang.state.time) and their associated significance values[92,93]. The dietary categories of herbivory and mycophagy were excluded from this analysis due to being represented either in only one partition

(mycophagy) or by only one species per partition (herbivory) (see Fig. 1).

## Evolutionary modelling

We fitted 12 competing models of evolution to assess differences in modes of evolution across diets and partitions ("Ameridelphia" and Australidelphia), using scores of the first four Principal Components of allometric space (76.12% of variance) and R package mvMORPH v. 1.1.6[94]. These models can be grouped in three broad categories; Early Burst (EB), Brownian Motion (BM) and Ornstein-Uhlenbeck (OU). EB is a model of evolution where most of the diversification occurred early on in the evolutionary history of the group, followed by a sharp decrease in disparity (usually interpreted as evidence of an adaptive radiation). BM models assume a random-walk process, where disparity increases linearly across evolutionary time. This model can be modified to assume a single or multiple evolutionary rates and a common or multiple ancestral states. Based on this, we fitted seven different BM models: a single-rate BM model (BM1), a multiple-rate BM model based on partitions (BMMs), a multiple-rate BM model based on diet (BMMd), a multiple-rate BM model based on a partition × diet interaction (BMMsd), a BM model with multiple ancestral states based on partition (BMMsm), a BM model with multiple ancestral states based on diet (BMMdm) and a BM model with multiple ancestral states based on a partition × diet interaction (BMMsdm). OU models assume evolution under stabilising selection towards a selective optimum. We fitted four OU models, one assuming a single optimum (OU1), one assuming different optima based on partition (OUs), one assuming different optima based on diet (OUd), and one assuming different optima based on a partition × diet interaction (OUsd). Evolutionary model fitting was implemented using R package Phytools and function *make.simmap*[95,96]. Model selection was based on sample-size corrected Akaike information criteria (AICc), with models with a delta (Δ) AICc below two considered as the best-supported models.

Plot displays were created with colours from R package RColorBrewer version 1.1.-3 (https://colorbrewer2.org/#). An overview of the pipeline for all analyses is presented in the Supplementary Material (Supplementary Fig. 2).

## Reporting summary

Further information on research design is available in the Nature Portfolio Reporting Summary linked to this article.

## Data availability

The data generated in this study have been deposited in the Zenodo database[86] at https://zenodo.org/record/7804164#.ZC5No_ZBxZc. The data generated in this study are provided in the Source Data file. Source data are provided as a Source Data file. Data were compiled from Flores et al.[41,77] and are available at https://link.springer.com/article/10.1007/s13127-018-0369-3#Sec16 and at https://link.springer.com/referenceworkentry/10.1007/978-3-030-88800-8_6-1#Sec18, Phylopic images are: https://www.phylopic.org/images/55e1c3e9-940c-486a-90d7-4b8e5663d248/thylacinus-cynocephalus, https://www.phylopic.org/images/73e9af73-e873-4aa8-90be-8d4a9afb8617/perameles-bougainville, Sarah Werning (unchanged) https://www.phylopic.org/images/dde4f926-c04c-47ef-a337-927ceb36e7ef/dromiciops-gliroides, https://www.phylopic.org/images/0e1fe113-feb7-46db-8a10-12f22f80332c/didelphis-virginiana, Sarah Werning (unchanged) https://www.phylopic.org/images/f34ca418-a0d9-4ed7-bc3c-a74bcdeae443/caenolestes-fuliginosus, https://www.phylopic.org/images/b62bab6e-99e9-4525-9b89-f5fb94742112/macropus-macropus. Source data are provided with this paper.

## Code availability

R code is deposited in Github and available at https://github.com/labw09/marsupialsallom.

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

## Acknowledgements

We thank the following curators, who provided permission for the examination of osteological material: Sergio Lucero of the Museo Argentino de Ciencias Naturales (Buenos Aires); Bruce Patterson and Bill Stanley of the Field Museum of Natural History (Chicago); Kris Helgen, Darrin Lunde, and Linda Gordon of the Smithsonian Institution (Washington, D.C.); Rob Voss and Eileen Westwig of the American Museum of Natural History (New York); Paula Jenkins of the British Natural History Museum (London); Sandy Ingleby of the Australian Museum (Sydney); Kenny Travouillon of the Western Australian Museum (Perth); Joao Alves de Oliveira of the Museu Nacional, Universidade Federal do Rio de Janeiro (Rio de Janeiro); and Mario de Vivo of the Museu do Zoologia Universidad de São Paulo (São Paulo). This research was supported by Australian Research Council grants FT200100822 awarded to L.A.B.W., DP180100792 awarded to M.A. and S.J.H, a University of Toronto Scarborough Postdoctoral Fellowship awarded to C.L.-A. and PICT funding (2015-238) awarded to N.P.G., D.F., and F.A. Data collection was additionally supported by funding from CONICET PIP (2021-23 11220200102778CO) and PICT (2016-3682) to N.P.G, and from PICT (2020-02042) of Agencia Nacional de Promoci on Cientıfica y Tecnica and PIP 928 of the CONICET to D.F. CONICET and the National Research Foundation of South Africa provided support to F.A.

## Author contributions

L.A.B.W. conceived of the study; L.A.B.W., F.A., D.F. and N.G. contributed to data collection; C.L.-A. and L.A.B.W. performed analyses; L.A.B.W. drafted the manuscript with review by S.H. and M.A., and input from all authors.

## Competing interests

The authors declare no competing interests.
