## [Peer Review File · Nature Communications]

Patterns of ontogenetic evolution across extant marsupials reflect different allometric pathways to ecomorphological diversityREVIEWER COMMENTS

Reviewer #1 (Remarks to the Author):

Wilson et al. examine patterns on ontogenetic allometry in the cranium of extant marsupials, specifically comparing ontogenetic trajectories (how shape changes with size during growth) among species in Australasia and the Americas. They show significant differences in slope and intercept between the 2 groups, as well as between most marsupial orders, dietary categories, and interactions between the two. The main exception is between animalivores and omnivores in the 2 regions, who share a similar ontogenetic allometric slope despite evolving independently, with animalivorous species appearing convergent. Despite Australasian marsupials being notably more speciose (~70% of all marsupials) and ecologically diverse than those in the Americas, variance in their cranial growth rates (here called allometric disparity) are not significantly higher.

Overall the methodology appears sound, although I have some comments below regarding sampling and presentation of the results.

Comments:

I don't normally comment on titles, however I find this one misleading in terms of 'across all extant marsupials'. While the sample size is not small (2091 specimens from 62 species in 18 families), it is certainly far from representative – or at least the authors don't tell us how the sampling was designed to represent extant morphological variation and/or body sizes. Furthermore tests of ontogenetic evolution are based on a reduced dataset of 50 species, which accounts for only ~14% of extant marsupial richness according to reference [9], or even less (12,5%) if going by the authors' statement of 400 living species. That's alright since I realise these growth series are hard to put together from museum collections, but I still think the authors should remove 'all' from the title.

Given that this study is based on allometry, I would expect more information on size variation in these groups and if/how that was considered during sampling. Especially for the more speciose orders like Dasyuromorphia, Didelphimorphia and Diprotodontia. Are the size axes across these groups reasonably represented? As in smallest, medium, largest? Because the species are never listed in the main text, it will be difficult for the reader to know, especially if they are not familiar with marsupials (Nature Communications is after all a broad journal). I would go so far as to ask that species names are shown in Figure 1 since there seems to be space (the tree diameter can be reduced without losing information), and to change the tip symbol sizes to be proportional to body mass, or use body mass bins (e.g., 0-1 kg, 1-5, 5-10, 10-20, etc.). That would give a more holistic overview of the dataset. Another, possibly better option is to show the phylogeny for all extant marsupials with proportional tip symbols and dietary categories (perhaps Orders can be labelled & colour coded along the outside rim with a representative silhouette for each, following the current design), but only write in species names for the sample. This would even better show how the study represents modern diversity, while being more transparent about what is included and orienting non-marsupial familiar readers. Because the sample is actually quite limited in terms of richness, species in this context are important enough to be shown in the main body, not just in the supplement. For 62 species, that should be possible.

L154 among marsupial orders (=8) – shouldn't this be 7?

L 161 pooled trajectories for dietary categories – should these be pooled if the species within them show significantly different slopes? This could be tested with a preliminary HOS test

L174/Fig 1 allometric space – it was not until I got to the Methods that I understood what this means, apparently a PCA on the allometric coefficients of the species cranial shape~size regressions for the 14 measured traits? I had to look at the R code to try to figure this out, and I'm still not quite sure if I have it right. The paper introducing this method [48] has been cited 68 times since its publication in 2008, so it appears to be an acceptable approach, however I'm still having a hard time wrapping my head around it. How are all the sampled individuals, i.e. growth series, captured in this species-level analysis? Sorry if I'm simply misunderstanding something,

but perhaps the authors could give a slightly longer description of the method and what it does. Lead author Wilson has used it 3 other times [7,58,59], so they are apparently experienced with the method.

L332 100 simulations seems rather low. I randomly checked 4 papers using this package and they all had a higher number: 200, 500, 1000. Is it clear that the results have reached some stationarity? Why the first 4 PCs used and not more to capture ~99% of the total variation?

L411 Nice to see a discussion of the 'outliers' in Fig 2B, but it would be even more helpful to label them in the graph, perhaps with some skull images. Likewise for both Fig 2A and B, it would be nice to have some indication of the actual shape changes occurring along the axes, like those mentioned starting on L186. Otherwise it's hard to know what these patterns mean morphologically.

L445 Sears 2004 should be [68]

L455 A long sentence starts here that is hard to follow. Please break up

L477 extreme feeding behaviour specialisations – please remind us what these are again (e.g.,)

L479 Calyuromyinae – perhaps add (wooly opossums and relatives)

Reviewer #2 (Remarks to the Author):

The authors have taken a novel approach to addressing the subject of unbalanced disparity among clades. Using the marsupial mammal cranium to assess morphological disparity, they test the hypothesis that the unbalanced disparity in adult South American species is lower than in Australian species due to less variation in their ontogenetic allometry trajectories. They also examine whether convergent dietary preferences arises due to convergent ontogenetic allometry trajectories, also a novel approach.

Overall, I enjoyed reading and reviewing this manuscript. The dataset is excellent and the authors should be commended for their efforts. A total of 2091 species from 62 species is excellent sampling and power to achieve their aims.

The order of the sections was a bit confusing, with methods coming after the results but before discussion. If the results are to be presented prior to the methods, then this section needs to be revised to provide enough comprehension, especially since there are many analytical components that are complex enough to require a flowchart in the supplementary materials.

My main concern was in the approach for the ontogenetic allometry patterns by clade, order and diet. As I read the introduction, I envisioned the possible outcomes and imagined a multivariate regression graph with a heap of linear trajectories, each a species, and tests for differences in their slopes within each group. Or using an allometric space from PCA of the trajectories (which was done), and analysis of variance on these trajectories. But instead from the results I read the species data were pooled. I don't get this information from the methods (281-299). However I gather from the models provided that all the specimens across species were pooled in those analyses, so the analysis was not actual species trajectories being considered with respect to clade, order or diet. This does not address the hypotheses in my understanding of them. Either the manuscript needs strong justification of why pooling can be done (are the species slopes all homogenous within a group?), or a different analysis needs to be done to test for variation of slopes within each group and between the groups.

The disparity analysis on the allometric space is answering the hypothesis about South Australian versus Australia, but the two groups have very different numbers of species. In disparity literature rarefaction analysis is considered necessary to deal with uneven groups, and it may be necessary

to provide this here, or justification for why it is not necessary.

The visual component of this manuscript is very limited and does not do the paper justice. In particular it is missing the raw trajectories (in a PCA space or multivariate regression space). From this it would be able to justify the pooling (though I am sceptical of this approach as it stands), and it helps the reader who is unfamiliar with the allometry space approach to understand how it is made. Furthermore, most of the results are provided in supplementary materials; graphical results in the main text is much more accessible and provides a stronger narrative since the reader can visualise the support for the hypotheses. Consider graphing the disparity values also to demonstrate the magnitude differences on ontogenetic diversity.

Minor comments

Line 322 here MV is used, but on line 196 PV is used.

Species names in the introduction are common names, while in the results and discussion the Latin names are used. Suggest for broader readership common names are used where possible (and without ambiguity).

Figure 1 orange and blue around the outside is very pale and hard to see. And quaternary and Neogene are not differentiated enough

Figure 2 suggest making the orange convex hull brighter as it is hard to see against the blue

RESPONSE TO REVIEWERS' COMMENTS

Reviewer #1 (Remarks to the Author):

Wilson et al. examine patterns on ontogenetic allometry in the cranium of extant marsupials, specifically comparing ontogenetic trajectories (how shape changes with size during growth) among species in Australasia and the Americas. They show significant differences in slope and intercept between the 2 groups, as well as between most marsupial orders, dietary categories, and interactions between the two. The main exception is between animalivores and omnivores in the 2 regions, who share a similar ontogenetic allometric slope despite evolving independently, with animalivorous species appearing convergent. Despite Australasian marsupials being notably more speciose (~70% of all marsupials) and ecologically diverse than those in the Americas, variance in their cranial growth rates (here called allometric disparity) are not significantly higher.

Overall the methodology appears sound, although I have some comments below regarding sampling and presentation of the results.

Comments:

I don't normally comment on titles, however I find this one misleading in terms of 'across all extant marsupials'. While the sample size is not small (2091 specimens from 62 species in 18 families), it is certainly far from representative – or at least the authors don't tell us how the sampling was designed to represent extant morphological variation and/or body sizes. Furthermore tests of ontogenetic evolution are based on a reduced dataset of 50 species, which accounts for only ~14% of extant marsupial richness according to reference [9], or even less (12,5%) if going by the authors' statement of 400 living species. That's alright since I realise these growth series are hard to put together from museum collections, but I still think the authors should remove 'all' from the title.

>done, we've removed 'all' from the title, as requested.

Given that this study is based on allometry, I would expect more information on size variation in these groups and if/how that was considered during sampling. Especially for the more speciose orders like Dasyuromorphia, Didelphimorphia and Diprotodontia. Are the size axes across these groups reasonably represented? As in smallest, medium, largest? Because the species are never listed in the main text, it will be difficult for the reader to know, especially if they are not familiar with marsupials (Nature Communications is after all a broad journal). I would go so far as to ask that species names are shown in Figure 1 since there seems to be space (the tree diameter can be reduced without losing information), and to change the tip symbol sizes to be proportional to body mass, or use body mass bins (e.g., 0-1 kg, 1-5, 5-10, 10-20, etc.). That would give a more holistic overview of the dataset. Another, possibly better option is to show the phylogeny for all extant marsupials with proportional tip symbols and dietary categories (perhaps Orders can be labelled & colour coded along the outside rim with a representative silhouette for each, following the current design), but only write in species names for the sample. This would even better show how the study represents modern diversity, while being more transparent about what is included and orienting non-marsupial familiar readers. Because the sample is actually quite limited in terms of richness, species in this context are important enough to be shown in the main body, not just in the supplement. For 62 species, that should be possible.

>Thanks for these suggestions. We've re-drawn Fig. 1 following your suggestion – we've added a scaled symbol at the tip for body mass, we've labelled groups and species names, and we've added silhouettes to illustrate these. Average adult body mass in the sample ranged over 5 orders of magnitude. In Dasyuromorphia, body mass ranged from 4 grams to 16.7 kg, in Didelphimorphia the range was 19 grams to 1.3 kg, and in Diprotodontia, this was 9 grams to 39 kg.

We've added information on body mass variation to the main text to highlight the diversity in size across the sample, please see lines 455-460.

L154 among marsupial orders (=8) – shouldn't this be 7?

>thanks, typo corrected

L 161 pooled trajectories for dietary categories – should these be pooled if the species within them show significantly different slopes? This could be tested with a preliminary HOS test

>thanks, we initially ran pairwise comparisons, these numbered 1109 comparisons, of which 131 showed differences in slope (11%). Given the high number of pairwise comparisons, and the high number of non-significant differences in slope, we chose to pool trajectories to enable results to be presented and digested. We've now added an explanation to the methods to this effect (line 448-454) and for clarity we've also provided the preliminary HOS tests in the supplementary material (Supplementary Table 1).

We've also now added the results and discussion of the species-level pairwise comparison results to the main text and we've also provided those raw data in supplement (please see Supporting Data File 1 and Supporting Data File 2). We've also graphed the raw trajectories in new Figure 2.

Please see lines 520-526 (Methods) and lines 155-168 (Results).

L174/Fig 1 allometric space – it was not until I got to the Methods that I understood what this means, apparently a PCA on the allometric coefficients of the species cranial shape~size regressions for the 14 measured traits? I had to look at the R code to try to figure this out, and I'm still not quite sure if I have it right. The paper introducing this method [48] has been cited 68 times since its publication in 2008, so it appears to be an acceptable approach, however I'm still having a hard time wrapping my head around it. How are all the sampled individuals, i.e. growth series, captured in this species-level analysis? Sorry if I'm simply misunderstanding something, but perhaps the authors could give a slightly longer description of the method and what it does. Lead author Wilson has used it 3 other times [7,58,59], so they are apparently experienced with the method.

>thanks, we're sorry that this method step wasn't described with enough detail. You're correct, species-level regressions of 14 measurements were run for all individuals of each species, producing 14 allometric coefficients per species, representing growth trends across all sampled ontogenetic specimens for that species. These were then input into the PCA, so each point in allometric space represents the allometric coefficients output from regression of shape ~ size for all specimens for that species. We've added an example to the methods text to illustrate this further (lines 539-)

*“For each species, the allometric coefficients represent the growth trends for each measurement, as assessed across all specimens in the sample. As an example, for the opossum *Caluromys philander* (represented by 40 specimens), the multivariate regression of shape variables ($n=14$) ~ size performed on 40 specimens, produces a set of 14 allometric coefficients, capturing the ontogenetic allometric trajectory. These 14 coefficients were compiled for each species ($n=50$) in the pruned sample ($n = 1949$ specimens) and used as input to the PCA (50 taxa \times 14 allometric coefficients).”*

L332 100 simulations seems rather low. I randomly checked 4 papers using this package and they all had a higher number: 200, 500, 1000. Is it clear that the results have reached some stationarity? Why the first 4 PCs used and not more to capture ~99% of the total variation?

> Thanks for raising this. We chose 4 PCs as the maximum possible for this simulation to execute on a workstation, after assessing the significance of variation captured by each principal component using a broken stick model (Supplementary Figure 1). We tried 100, 200, 300 and 1000 simulations, and found that >200 (24 hours compute time on a 64GB RAM desktop) did not yield differences in C value >0.0003, which indicates stationarity to us. We've updated our results to present the simulations for 300 runs (Supplementary Table 10), and have added justification to the text, please see lines 568-574.

L411 Nice to see a discussion of the ‘outliers’ in Fig 2B, but it would be even more helpful to label them in the graph, perhaps with some skull images. Likewise for both Fig 2A and B, it would be nice to have some indication of the actual shape changes occurring along the axes, like those mentioned starting on L186. Otherwise it’s hard to know what these patterns mean morphologically.

> Done. We’ve redrawn Figure 2 (now 3) and we have added to the species names directly to the plot. We’ve also added text to describe the growth changes. Please see new Figure 3.

L445 Sears 2004 should be [68]

>thanks, corrected

L455 A long sentence starts here that is hard to follow. Please break up

>done, we’ve split this into two sentences.

L477 extreme feeding behaviour specialisations – please remind us what these are again (e.g., ...)

>done, we’ve added “(e.g., myrmecophagy)”

L479 Calyuromyinae – perhaps add (wooly opossums and relatives)

>done, we’ve added the suggested text

Reviewer #2 (Remarks to the Author):

The authors have taken a novel approach to addressing the subject of unbalanced disparity among clades. Using the marsupial mammal cranium to assess morphological disparity, they test the hypothesis that the unbalanced disparity in adult South American species is lower than in Australian species due to less variation in their ontogenetic allometry trajectories. They also examine whether convergent dietary preferences arises due to convergent ontogenetic allometry trajectories, also a novel approach.

Overall, I enjoyed reading and reviewing this manuscript. The dataset is excellent and the authors should be commended for their efforts. A total of 2091 species from 62 species is excellent sampling and power to achieve their aims.

The order of the sections was a bit confusing, with methods coming after the results but before discussion. If the results are to be presented prior to the methods, then this section needs to be revised to provide enough comprehension, especially since there are many analytical components that are complex enough to require a flowchart in the supplementary materials.

>Apologies for this confusion. We’ve updated the main sections to now following the order

Abstract

Introduction

Results

Discussion

Methods

>Since the Methods need to be presented after the Results per journal format, we’ve add a descriptive sentence at the start of each results sub-section to provide more context to each result, following the reviewer’s request.

My main concern was in the approach for the ontogenetic allometry patterns by clade, order and diet. As I read the introduction, I envisioned the possible outcomes and imagined a multivariate regression graph with a heap of linear trajectories, each a species, and tests for differences in their slopes within each group. Or using an allometric space from PCA of the trajectories (which was done), and analysis of variance on these trajectories. But instead from the results I read the species data were pooled. I don’t get this information from the methods (281-299). However I gather from the models provided that all the specimens across species were pooled in those analyses, so the analysis was not actual species trajectories being considered with respect to clade, order or diet. This does not address the hypotheses in my understanding of them. Either the manuscript needs strong justification of why

pooling can be done (are the species slopes all homogenous within a group?), or a different analysis needs to be done to test for variation of slopes within each group and between the groups.

>Thanks for this comment, we did initially run HOS slope tests at a pairwise level across all species (1109 comparisons), that yielded 131 significant differences (11%). Due to the low number of significant differences and the sheer volume of comparisons and variables involved (partition, diet, Order), we decided to present the pooled results.

We have now graphed the raw slopes – please see new Figure 2, as requested. We have also added in the results for the variation in slopes at species-level to the main text and discussion. We've also provided the raw results for variation of slopes within each group, please see Supporting Data Files 1 and 2.

The disparity analysis on the allometric space is answering the hypothesis about South Australian versus Australia, but the two groups have very different numbers of species. In disparity literature rarefaction analysis is considered necessary to deal with uneven groups, and it may be necessary to provide this here, or justification for why it is not necessary.

>Thanks, we agree. Morphological disparity was calculated using Procrustes Variance (PV), which is the sum of the diagonal elements of the group covariance matrix divided by the number of observations in the group. Therefore, the PV values are scaled according to group size prior to statistical comparison of groups, already accounting for unequal group sizes. We have edited the methods section to clarify that this was already done, please see line 322-324.

The visual component of this manuscript is very limited and does not do the paper justice. In particular it is missing the raw trajectories (in a PCA space or multivariate regression space). From this it would be able to justify the pooling (though I am sceptical of this approach as it stands), and it helps the reader who is unfamiliar with the allometry space approach to understand how it is made. Furthermore, most of the results are provided in supplementary materials; graphical results in the main text is much more accessible and provides a stronger narrative since the reader can visualise the support for the hypotheses. Consider graphing the disparity values also to demonstrate the magnitude differences on ontogenetic diversity.

>done. We've added two new figures to improve the visualisation of the results. Figure 2 (new) provides the raw ontogenetic trajectories, and Figure 4 (new) provides the disparity values, by dietary group and phylogenetic group. We've also edited Figure 1 and 3 to make both of these more visually informative. We've also included the results of the within-group trajectory analysis (i.e., the non-pooled data) as per your above comment.

Minor comments

Line 322 here MV is used, but on line 196 PV is used.

>thanks we've corrected this to PV as per the rest of the manuscript.

Species names in the introduction are common names, while in the results and discussion the Latin names are used. Suggest for broader readership common names are used where possible (and without ambiguity).

>done, we've included common names throughout now.

Figure 1 orange and blue around the outside is very pale and hard to see. And quaternary and Neogene are not differentiated enough

>thanks, we've re-drawn Fig 1 completely now, to incorporate changes suggested by reviewer #2, and have labelled the time periods, as well as enhanced the contrast for orange/blue.

Figure 2 suggest making the orange convex hull brighter as it is hard to see against the blue

>done. We've redrawn Figure 2 (now Figure 3), also incorporating changes requested by reviewer #1.

** See Nature Portfolio's author and referees' website at www.nature.com/authors for information about policies, services and author benefits.

REVIEWERS' COMMENTS

Reviewer #1 (Remarks to the Author):

I am satisfied with the revised version and find the manuscript much improved with the requested changes and additional figures. I only found one typo in the new text - L 276 should be 'less morphologically diverse'.

Well done to the authors.

Reviewer #2 (Remarks to the Author):

The revisioned manuscript, particularly the updated visualisations, have greatly improved the readability of the manuscript. I have no further comments to make, and thanks the authors for the dilligent revisions.

RESPONSE TO REVIEWERS' COMMENTS

Reviewer #1 (Remarks to the Author):

I am satisfied with the revised version and find the manuscript much improved with the requested changes and additional figures. I only found one typo in the new text - L 276 should be 'less morphologically diverse'.

Well done to the authors.

>thanks, we've corrected the typo.

Reviewer #2 (Remarks to the Author):

The revisioned manuscript, particularly the updated visualisations, have greatly improved the readability of the manuscript. I have no further comments to make, and thanks the authors for the dilligent revisions.

>thanks